# *Plasmodium chabaudi* Infection Alters Intestinal Morphology and Mucosal Innate Immunity in Moderately Malnourished Mice

**DOI:** 10.3390/nu13030913

**Published:** 2021-03-11

**Authors:** Noah Joseph Murr, Tyler B. Olender, Margaret R. Smith, Amari S. Smith, Jennifer Pilotos, Lyndsay B. Richard, Chishimba Nathan Mowa, Michael Makokha Opata

**Affiliations:** Department of Biology, College of Arts and Sciences, Appalachian State University, 572 Rivers St, Boone, NC 28608, USA; murrnj@appstate.edu (N.J.M.); olendertb@appstate.edu (T.B.O.); margsmit@wakehealth.edu (M.R.S.); smithas1@appstate.edu (A.S.S.); pilotosjenni@gmail.com (J.P.); richardly20@ecu.edu (L.B.R.); mowacn@appstate.edu (C.N.M.)

**Keywords:** malnutrition, malaria, gut immunity, intestinal permeability, innate immunity

## Abstract

*Plasmodium falciparum* is a protozoan parasite which causes malarial disease in humans. Infections commonly occur in sub-Saharan Africa, a region with high rates of inadequate nutrient consumption resulting in malnutrition. The complex relationship between malaria and malnutrition and their effects on gut immunity and physiology are poorly understood. Here, we investigated the effect of malaria infection in the guts of moderately malnourished mice. We utilized a well-established low protein diet that is deficient in zinc and iron to induce moderate malnutrition and investigated mucosal tissue phenotype, permeability, and innate immune response in the gut. We observed that the infected moderately malnourished mice had lower parasite burden at the peak of infection, but damaged mucosal epithelial cells and high levels of FITC-Dextran concentration in the blood serum, indicating increased intestinal permeability. The small intestine in the moderately malnourished mice were also shorter after infection with malaria. This was accompanied with lower numbers of CD11b^+^ macrophages, CD11b^+^CD11c^+^ myeloid cells, and CD11c^+^ dendritic cells in large intestine. Despite the lower number of innate immune cells, macrophages in the moderately malnourished mice were highly activated as determined by MHCII expression and increased IFNγ production in the small intestine. Thus, our data suggest that malaria infection may exacerbate some of the abnormalities in the gut induced by moderate malnutrition.

## 1. Introduction

Macro and micronutrient deficiencies lead to malnutrition which can significantly affect disease outcomes. It has been shown that children suffering from protein energy malnutrition (PEM) in malaria endemic areas tend to have lower parasite burden during infection, which is typically due to glucose-6-phosphate dehydrogenase deficiency [1,2]. Despite lower parasite loads in this patient population, historically, research has indicated that children with PEM have increased morbidity in response to infections [3]. Malnutrition due to PEM is associated with cerebral malaria, which is the severe form of the disease that leads to death or neurological deficits after recovery from malaria infection [4]. With more than half a million childhood deaths reported in sub-Saharan Africa being caused by diarrhea, malaria, and malnutrition [5], it is critically important to investigate the association between malaria and malnutrition and how this affects gut immunity.

*Plasmodium* infections, which cause malaria disease, lead to symptoms such as headache, arthralgia, and abdominal discomfort including anorexia, nausea, pain, vomiting and diarrhea [6]. In a west-African study to determine the effect of zinc on malaria infection, zinc supplementation significantly reduced the rate of diarrhea in children infected with malaria. This intervention reduced health center visits due to malaria disease by over 30% compared to those receiving placebo [7,8]. Thus, zinc supplementation led to better disease outcomes resulting in lower hospital visits. Similar observations have been reported by other investigators where supplementation of zinc and vitamin A reduced complicated malaria diagnosis in children by 27% [9].

Iron deficiency is another common nutrient deficiency in African children and malaria infection causes anemia as the parasites deplete red blood cells [10,11]. High iron is associated with increased risk of malaria incidence [12], and one study has shown that iron deficiency may be protective against malaria [13]. Despite this, more recent studies have shown the benefits of iron supplementation when used in conjunction with malaria prevention strategies [14]. Due to prevalence of iron deficiency in malaria endemic areas and the effects on cognitive development, the World Health Organization (WHO) recommends iron supplementation in conjunction with effective malaria prevention and regimens in these regions [15]. In a recent study to investigate whether iron status influences risk of malaria infection in children, it was found that iron deficiency may protect against malaria infection when defined using ferritin and transferrin saturation, but not when defined by hepcidin, soluble transferrin receptors, and hemoglobin [16].

The innate immune system is an integral part of the response to malaria infection. This is especially true when the *Plasmodium* parasite enters the symptomatic stage and infects red blood cells [17]. An optimal innate immune response to this stage of infection is critical to host survival, without it the host will potentially succumb to infection [18]. Dendritic cells and macrophages are the first and perhaps most important responders during this stage, charged with controlling parasite growth via production of cytokines and communication with the adaptive immune response through antigen presentation [19]. More specifically, dendritic cells and macrophages expressing CD11c and CD11b respectively, have been shown to be imperative in controlling the initial infection and promoting host survival [20,21].

Even though the importance of CD11c^+^ and CD11b^+^ cells during *Plasmodium* infection is well established [20,21,22], it is unclear how the activation status of these innate immune cells change during *Plasmodium chabaudi* (*P. chabaudi)* infection in the gut tissues. Indeed, malaria is common in areas with episodes of malnutrition [23]. Recently, immunity in the gut and the relationship with the microbiome has driven researchers to investigate the effects of diet and disease on the gut [24,25,26,27]. It has been shown that poor nutritional status may drive dysbiosis in microbial communities, increase intestinal permeability (IP), and negatively impact mucosal innate immunity [28]. 

Despite the wealth of knowledge available regarding malaria and malnutrition, the dynamic relationship between the two has resulted in conflicting conclusions on the potentially protective or exacerbative effects that malnutrition has on malarial disease [29,30,31,32]. More specifically, confounding conclusions about malaria and malnutrition led us to study how *Plasmodium* infection alters mucosal innate immunity during moderate malnutrition as malaria alone has been reported to cause intestinal dysbiosis in healthy animals [33].

In this study, we used a moderately malnourished diet comprised of 3% protein that is deficient in zinc and iron to investigate the effects of moderate malnutrition on gut architecture and immunity. This diet closely models moderate malnutrition in parasite endemic regions as reported by other investigators [34]. We used the *P. chabaudi* rodent parasite strain that induces disease manifestations like the *P. falciparum* seen in humans. Using this model, we were able to demonstrate that moderate malnutrition and malaria infection alters intestinal architecture and mucosal innate immunity in mice. These results serve as a valuable indicator for disease morbidity and progression in humans.

## 2. Materials and Methods

### 2.1. Mice and Parasite

Adult C57BL/6 mice were obtained from Harlan labs and a breeding colony maintained in our animal facility. The rodent strain of malaria, *P. chabaudi* was received as a gift from Dr. Robin Stephens at the University of Texas Medical Branch Galveston. Authorization to use the parasite was given by Dr. Jean Langhorne from the Francis and Crick Institute, UK. All animal studies are approved by the Institutional Animal Care and Use Committee (IACUC) at Appalachian State University. Male mice aged 8–14 weeks were used for experiments with consistency in both the malnourished and control groups.

### 2.2. Malnutrition and Infection

Mice were fed either a moderately malnourished diet (Mal; TD.99075) with 3% protein and deficient in iron and zinc, or a well-nourished diet (Ctrl; TD.99103) with 17% protein and all necessary micronutrients from Envigo/Teklad (Indianapolis, IN). Both diets have similar caloric content, which is made up by extra carbohydrates in the moderately malnourished diet. The diets were administered to each group of mice at 3 g per mouse, for 3 mice per group, based on our approve protocol. We limited the diet to 3 g per mouse so that they did not overeat to compensate for nutrient loss. The mice were fed daily for approximately 4–5 weeks to induce moderate malnutrition until sacrifice. After 4–5 weeks, both the control and malnourished mice were infected with 1 × 10^5^
*P. chabaudi* intraperitoneally and other groups of control and malnourished mice were left uninfected to serve as negative controls. At 9 days post-infection, all mice were culled via cervical dislocation with adherence to our approved IACUC protocol 20–10. The gut tissues including, the small and large intestine and cecum were harvested and placed in ice cold PBS supplemented with 2% FBS and 0.02% EDTA (Atlanta biologicals S11150H, Flowery Branch, GA, USA) in preparation for cleaning.

### 2.3. Physiology of Malaria and Malnutrition

To determine parasitemia, blood smears were collected at days 3, 5, 7, and 9 days post-infection via tail snips. Slides were stained with Camco Quik Stain II (Sigma-Aldrich, St. Louis, MO, USA) for 10 s and rinsed with DiH_2_O for 20 s. Parasites were then counted on a standard compound light microscope and parasitemia was calculated as percent infected red blood cells. Temperatures were taken every day in the infected mice using a rectal probe and thermometer (Braintree Scientific, Braintree, MA, USA). Body weights were also recorded every day after infection.

### 2.4. Preparation of Gut Tissues for Flow Cytometry

The gut tissues were cleaned by manually removing all residual adipose residues with scissors and forceps. The tissues were then cut longitudinally and flushed with ice cold PBS supplemented with 0.02% EDTA using a 21-gauge needle and 10-mL syringe to remove fecal matter. Any residual fecal matter was removed by scraping with forceps and rinsing with more ice-cold PBS + 0.02% EDTA buffer. The tissues were then placed into 6-well plates with ISCOVES culture media (Corning #10-016-CV) supplemented with 2 mM L-glutamine (Atlanta biologicals B21210), 5 mM Sodium Pyruvate (Gibco 11360-070), non-essential amino acids (MEM NEAA) (Gibco 11140-050), 10 mM HEPES (Gibco 15630-080), 100 U/mL Penicillin, 100 U/mL Streptomycin and 2 × 10^−5^ M of 2β-Mercaptoethanol (Gibco 21985-023) then minced with scissors. Type I collagenase (ThermoFisher #17018029) was added at a concentration of 100 U/mL, followed by a one-hour incubation at 37 °C and 5% CO_2_ for extraction of lamina propria cells. Cells were agitated every 15 min to ensure homogeneity.

### 2.5. Flow Cytometry (Surface Staining)

After one-hour incubation, the 6-well plates were removed from the incubator and the solid tissues were mashed through 70-μm nylon filters and lysed with 1× RBC lysis buffer for 1 min at room temperature. The RBC lysis was stopped by adding 5 mL ice-cold PBS and spun at 300× *g* at 4 °C for 5 min. The suspensions were then placed on ice in 5-mL round bottom tubes and cells were counted at a 1:10 dilution with trypan exclusion on a hemocytometer. After determining cell numbers, an aliquot of cells was taken and resuspended in cold FACS buffer (PBS, 2% FBS, and 0.1% NaN_3_ sodium azide) in new 5-mL round-bottom polystyrene tubes.

The cells were incubated with Fc block in the dark at 4 °C for 20 min to ensure specific binding. Fluorescent antibodies were used to label cells of interest: PE-Cy5-conjugated anti-CD11b (Tonbo 55-0112-U100), FITC-conjugated anti-CD11c (Tonbo 35-0114-U500), and PE-conjugated anti-MHCII (Tonbo 50-5321-U100) at 4 °C in the dark for 40 min. Fluorescently stained cells were washed in FACS buffer, resuspended in 300 μL of FACS buffer, filtered and collected on an FC500 flow cytometer (Beckman Coulter, Indianapolis, IN, USA).

### 2.6. Flow Cytometry (Intracellular Cytokine Staining)

Aliquots of cells were transferred into a sterile 24-well plate with 1 mL of complete ISCOVES culture media supplemented with 2 mM L-glutamine, 5 mM Sodium Pyruvate, non-essential amino acids (MEM NEAA), 10 mM HEPES, 100 U/mL Penicillin, 100 U/mL Streptomycin and 2 × 10^−5^M of 2β-Mercaptoethanol. The cells were stimulated in vitro with 1 μL of cell stimulation cocktail (Tonbo Biosciences, San Diego, CA, USA). The plate was then placed into a HERAcell 150i incubator set at 37 °C and 5% CO_2_ for 6 h. After incubation, the cells were harvested and spun at 300× *g* at 4 °C for 5 min.

The cells were washed with FACS buffer then incubated with Fc block in the dark at 4 °C for 20 min. Fluorescent antibodies were used against surface markers to label cells of interest: PE-Cy5-conjugated anti-CD11b (Tonbo 55-0112-U100) and PE-conjugated F4/80 (Biolegend, 123110) at 4 °C in the dark for 40 min. After incubation, cells were fixed with 300 μL of 2% paraformaldehyde. The cells were then permeabilized using permeabilization buffer (Tonbo Biosciences, San Diego, CA, USA) and stained with IFNγ (Biolegend, 505850). The cells were then resuspended with 200 μL of FACS buffer and filtered for analysis on the FC500 flow cytometer.

### 2.7. Assessment of Gut Pathology and Leakage 

Gut tissues were excised from d9 *P. chabaudi* infected mice, 3 mm sections of small intestine were cut and cleaned then fixed in 10% phosphate-buffered formalin. Tissues were fixed for approximately 48 h and embedded in paraffin. Sections 8 μm thick were cut using a microtome and stained with hematoxylin and eosin (Sigma-Alrich, St. Louis, MO, USA). Sections were viewed under an Olympus IX81/DP80 in the microscopy facility.

Intestinal permeability and gut leakage were studied using fluorescein isothiocyanate (FITC)-labeled dextran on day 9 post-infection. Mice were fasted overnight and weighed the next morning. FITC-dextran was prepared at a concentration of 100 mg/mL in sterile PBS and administered to each mouse at 4 mg/g body weight via oral gavage. Four hours after dextran administration, the mice were culled, and 0.5 mL of blood was taken via cardiac puncture. Blood samples were then spun down at 1677× *g* for 5 min. The sera were collected and diluted at a 1:1 ratio with sterile PBS. Dextran absorbance was read at 488 nm excitation/530 nm emission on a spectrophotometer. FITC-dextran concentrations in the sera were determined based on the standard curve.

### 2.8. Cytokine Determination by ELISA 

Blood obtained at euthanasia by cardiac puncher was clotted and centrifuged at 400× *g* to obtain sera which were frozen in aliquots at −80 °C for subsequent analysis. High affinity binding 96-well ELISA plates were coated with 100 μL of IFNγ and TNFα (0.5 mg/mL) capture antibody then incubated overnight at 4 °C. The plates were washed three times with PBS/Tween buffer and blocked by adding 200 μL of blocking solution to each well followed by a one-hour incubation at room temperature. The plates were washed again, and serum samples diluted at 1:2 dilution in PBS were added to respective wells, together with appropriate standards for each ELISA plate. The samples were then incubated at 4 °C overnight. The plates were then washed three times followed by addition of 100 μL of a 0.25 mg/mL biotin-labeled antibody diluted in the blocking solution and a one-hour incubation at room temperature. The plates were washed 3 times followed by addition of 100 μL of Av-Horse Radish Peroxidase and 30 min incubation. The plates were then washed five times and 100 μL of TMB substrate was added. The reaction was stopped after 10 min of incubation by adding TMB stop solution and the optical density was measured using a plate reader at 450 nm wavelength within 5 min of stopping the reaction.

### 2.9. Data Analysis

Raw data collected from the FC500 Beckman Coulter flow cytometer were analyzed using FlowJo software (Ashland, OR. Version 10.5.3). Calculations for all raw data were performed in Microsoft Excel (2008, Build 13127.21216) and graphs and statistics on the data were performed in Prism GraphPad Version 9.02 (San Diego, CA, USA). Statistical differences were determined using Two-way-ANOVA followed by Tukey’s test where appropriate. Data with a *p* value less than 0.05 were considered significantly different.

## 3. Results

### 3.1. Moderate Malnutrition Does Not Exacerbate Malaria Induced Pathology

To mimic a *Plasmodium falciparum* infection in malnourished humans, we infected male C57BL/6 mice with 1 × 10^5^
*P. chabaudi* that were fed a control diet or a moderately malnourished diet for 4 to 5 weeks. This enabled us to look at the effects of infection on gut mucosal innate immunity and leakage in a moderately malnourished environment. Using this model, we first looked at the effects of malaria infection and moderate malnutrition on parasitemia, body temperature, and percent weight change. The moderately malnourished mice had lower percent parasitemia when compared to well-nourished mice at day 9 post-infection, which is considered the peak of infection (POI), in this model (Figure 1A). As malaria infection is associated with high fever in humans or a decrease in temperature in rodent models, we determined changes in body temperature for the infected moderately malnourished and well-nourished mice. While both groups had similar trends in body temperature change for the first six days, the well-nourished mice dropped temperatures faster than the malnourished mice between days 7 and 8 post-infection (Figure 1B). This may be attributed to an exponential growth in parasitemia in this group which was significantly higher by day 9 p.i. (Figure 1A). When we looked at changes in body weight, the uninfected moderately malnourished mice incurred a significant percent body weight loss compared to the uninfected well-nourished mice. Upon infection, there was no significant difference in percent weight change between the malnourished and well-nourished groups, indicating that this malnutrition did not significantly affect disease pathology (Figure 1C). 

### 3.2. Changes in Gut Phenotype

Because malaria infection has been reported to affect mucosal integrity as reported by other investigators [33], we wondered if moderate malnutrition would result in increased damage to the gut mucosal surfaces. Thus, we evaluated the combined effects of *P. chabaudi* infection and moderate malnutrition on the physical appearance and leakage of the gut at the POI. Macroscopic analysis showed a slight yellowing of the small intestine in the moderately malnourished mice before infection, suggesting less luminal content and possible damage (Figure 2A, far right image). Upon infection the moderately malnourished mice typically had more yellowing indicating lymphangiectasia when compared to the malaria infected control well-nourished mice (Figure 2A). When we measured the length of each gut section to determine if the discoloration had an impact on gut lengths, the small intestine was significantly shorter after infection, compared to the uninfected or the infected well-nourished control (Figure 2B, left graph). However, there were no significant differences between all the groups in the lengths of the large intestine and cecum (Figure 2B).

We next investigated if the discoloration and shortening of the small intestine upon infection was due to damages in the mucosal architecture. To this end, we sectioned the small intestine and stained with H&E to check the epithelial cell structure and morphology. Consistent with discoloration in the uninfected moderately malnourished mice, we observed signs of epithelial cell deformities. This was significantly exacerbated in the infected malnourished mice (Figure 2C). To confirm if the deformed architecture of the mucosal epithelia was associated with gut leakage, we administered FITC conjugated dextran to all the experimental groups of mice via oral gavage after a 16-h fasting period. We collected serum 4-h after administrating dextran to determine leakage into the blood. As expected, we observed that moderate malnutrition alone without infection slightly increased FITC-dextran in the sera (Figure 2D). Similar to observations reported by others, well-nourished mice that are infected had increased FITC-dextran in the sera as well. This effect was exacerbated in the moderately malnourished mice that are infected with *P. chabaudi*, which further increased FITC-dextran concentrations in the sera (Figure 2D). The increased FITC-Dextran in the moderately malnourished mice after infection indicate a heavily compromised intestinal mucosa. Taken together, these results suggest mucosal damage in the moderately malnourished mice after malaria infection leading to gut leakage.

### 3.3. Moderate Malnutrition Alters Mucosal Innate Immunity

Since innate immunity is essential for mucosal gut sampling and it has been shown that microbial communities in the gut can modulate the severity of malaria infection [35], we reasoned that the paleness, shortening and deformities of the guts in the moderately malnourished mice could be due to alterations in innate immune cells or possible inflammation. To this end, we determined the effects of *P. chabaudi* infection during moderate malnutrition on mucosal innate immune cells in the gut. Flow cytometric analysis of lamina propria cell subsets; CD11b^+^CD11c^−^ macrophages, CD11b^+^CD11c^+^ myeloid dendritic cells, and CD11b^-^CD11c^+^ lymphoid dendritic cells [36,37,38,39,40], indicated that malnutrition alone did not have an effect on these cell populations. Upon infection, there were significantly lower numbers of the macrophage population in the large intestine of the malnourished mice. While there were trends towards lower cell numbers in the small intestine and cecum in the moderately malnourished mice after infection, these differences were not statistically significant (Figure 3A–C). The lower number of innate immune cells present at the POI in the moderately malnourished mice indicate a reduced immune response [41].

To further investigate if the lower number of innate immune cells in the moderately malnourished mice was associated with decreased functionality, we determined the proportion and number of CD11b^+^ macrophages expressing MHC II, a marker for activated cells capable of presenting antigen upon phagocytosis [42]. In the small intestine and cecum, moderate malnutrition alone did not affect activation of the macrophages. Upon infection, there was a slight increase in the number of macrophage cells expressing MHC II in the moderately malnourished mice, even though not statistically significant (Figure 4A,C). Interestingly, we found that the moderately malnourished mice had increased proportion and numbers of activated macrophages in their large intestine before and after infection (Figure 4B). The myeloid and dendritic cells were not different between the groups in all tissues (data not shown). Taken together with Figure 3, our data suggest that moderate malnutrition decreases the number of innate immune cells and the lower number of cells in these mice may be highly activated. 

### 3.4. Moderate Malnutrition Increases Cytokine Production by Macrophages in the Small Intestine during Malaria Infection

Based on trends showing an increase in CD11b+ macrophages expressing MHC II in the small intestine, we hypothesized that the macrophages would be producing more proinflammatory cytokines leading to discoloration and damage seen in Figure 2. Therefore, we investigated the proportion and number of cytokine secreting macrophages in the small intestine in both the moderately malnourished and well-nourished mice (Figure 5A). Cytokine production correlated with activation status with moderately malnourished mice showing significantly high numbers of IFNγ producing macrophages in the small intestine (Figure 5A, bottom panel). We wondered if this inflammatory cytokine production was localized or systemic. To investigate this, we examined the concentrations of inflammatory cytokines (IFN-γ and TNF-α) in the serum using ELISA. While there was no significance difference, the results indicated trends towards an increase in both TNF-α and IFN-γ in the moderately malnourished mice when compared to the well-nourished mice (Figure 5B). Taken together, these results suggest that the reduced number of macrophages during moderate malnutrition may be highly activated, thus increasing localized IFN-γ secretion and other pro-inflammatory cytokines that may lead to mucosal surface damage.

## 4. Discussion

Malaria and malnutrition have been occurring in sub-Saharan Africa for years [43,44,45]. Despite this, little is known about the detrimental effects of malaria infection on gut immunity during moderate malnutrition. Here, we have demonstrated that the number of mucosal innate immune cells, including early macrophage and dendritic cell progenitors, in the gut tissues are lower after *P. chabaudi* infection during moderate malnutrition. Even though there are diminished numbers of CD11b^+^ and CD11c^+^ cells in the moderately malnourished mice after infection, there is possible compensatory response by the CD11b^+^MHC II^+^ macrophages in both proportion and number in response to the infection. The highly activated macrophages produce more IFNγ that could induce inflammation and possible luminal damage, resulting in gut leakage as shown by high FITC-dextran concentrations in the sera of the moderately malnourished mice.

Malnutrition combined with *Plasmodium* infection contribute to immune-induced severe malarial anemia (SAM) [46], and increased metabolic acidosis, which can dramatically reduce appetite and result in weight loss seen at the peak of infection (POI) in the moderately malnourished mice in our study [47]. Similarly, ablations to gut epithelial cells in the small intestine of the moderately malnourished mice, which was increased with *Plasmodium* infection leading to a damaged mucosa may contribute to shortening of the tissue and a reduction in weight. As increased parasite load correlates with lower body temperature in mice, moderately malnourished mice did not lose as much weight as the well-nourished controls, correlating with lower parasite loads in the moderately malnourished group.

In a study by the Melby group using the same moderate malnutrition model as our project, they showed that malnutrition related reduction in macrophages and monocytes was associated with reduced lymph node-resident cells, but not migratory cells in response to infection [48]. Earlier studies by Fuss et al., reported that hypoproteinemia could lead to lymphangiectasia as a result of decreased presence of immune cells in the gut. This condition is caused by inhibitory effects to CD45RA^+^ cells migrating to the intestines [49]. The guts of the moderately malnourished mice exhibited macroscopic symptoms of lymphangiectasia, as seen by the paleness of the gut. This could be due to the high expression of CD11b^+^MHC II^+^ cells which lead to damage in the small intestine, hence altering immune cell migration to the gut mucosal surface. Other forms of micronutrient deficiency have shown similar compensatory response by activated macrophages [50,51]. This compensatory response is associated with increased inflammation, which is meant to protect the gut against pathogenic organisms, as well as diet induced dysbiosis via upregulation of MHC II [52,53]. It is highly possible that regulatory mechanisms are deformed. It has been reported that nutritional factors can modulate Treg plasticity, metabolism, and function [54], even though we did not test for Tregs and anti-inflammatory responses, as our study focused on innate immune cells.

In addition to altering regulatory mechanisms in the gut, the compensatory response due to malnutrition can have consequences regarding the integrity of the gut. Kim et al., showed that dysbiosis in the intestines can polarize macrophages towards an inflammatory phenotype, which is supported by malnutrition linked dysbiosis shown by Kumar et al. [55,56]. With increased dysbiosis due to malnutrition comes the exacerbated and damaging effect induced by malaria. Taniguchi and colleagues also showed that malaria infection in mice also causes dysbiosis and inflammation [30]. Therefore, malaria and malnutrition together can drive macrophages towards an inflammatory phenotype that is characterized by increased MHC II expression and cytokine secretion.

The increased proportion and number of CD11b^+^MHC II^+^ macrophages were seen in all sections of the gut, with significant differences in the large intestine. This result is further supported with a study by Stough et al., which highlighted the interaction between the microbiome and malaria infection with emphasis on the large intestine, the home of the microbiome [57]. Malnourished individuals experiencing dysbiosis have higher levels of inflammation, and their microbiomes are more susceptible to opportunistic pathogens [58]. Activated macrophages release inflammatory cytokines, which could cause increased intestinal permeability and shortening of the small intestine [42,59]. This could provide a possible explanation for the increased macrophage activation seen in the large intestine, not only during infection, but without infection as well in the moderately malnourished mice. The increased macrophage activation compounded by malnutrition could also cause physical damage to the gut leading to gut leakage [60,61,62]. Furthermore, diets deficient in protein have been found to induce pathological lymphangiectasia, which can directly affect lymphatic vessels and cause yellow discoloration due to chyle buildup [63].

To investigate this further we utilized intracellular cytokine staining, specifically looking at IFNγ secretion by CD11b^+^F4/80^+^ macrophages in the small intestine. In chronic infections such as malaria, elevated levels of IFNγ are seen locally in the gut [64,65]. The high inflammatory response by IFNγ negatively impact epithelial barrier in the mucosa [66]. More specifically, it has been associated with decreased expression of tight junction proteins, zonula occludins (ZO)-1, claudins, and occludin [67]. Furthermore, this effect also allows for dissemination of potentially pathogenic bacteria across the gut epithelial layer [68], a phenotype we plan so study using a severe malnutrition model. While we see epithelial cell ablation, and dextran leakage from the gut with moderate malnutrition, we do not think there is bacterial translocation in this model, based on lower systemic IFNγ and TNFα observed in the sera.

A commonly used method to test intestinal permeability is through the administration of FITC-Dextran, which does not normally pass through the gut epithelium [69]. Previous research by other groups showed that malaria and malnutrition can independently increase sera FITC-dextran concentrations [70]. The increased concentrations seen in our study support the hypothesis that malnutrition exacerbates leakage induced by malaria disease, as malaria infection alone leads to deformities in the gut epithelia, leading to gut leakage. To our knowledge, this is the first study to demonstrate the exacerbated effects of moderate malnutrition on gut epithelia during malaria infection. 

Our observations support studies by Singh et al., where they showed that unwarranted inflammation results from altered nutrition leading to increased intestinal permeability, which was also associated with shortening of the gut tissues [71]. In a different study by Burke et al., the researchers showed that intestinal shortening was typically due to excessive chronic inflammation that disrupts the epithelial and mucosal barriers [72]. To this effect, there is excessive buildup of the extracellular matrix and fibrosis of the tissue, resulting in shortening. This effect has also been seen in DSS-induced colitis in mice, which is characterized by chronic inflammation [73]. 

Male mice typically experience the highest amount of morbidity following infection, characterized by severe hypothermia, parasitemia, anemia, and weight loss [74]. Thus, it provided a better model for understanding the impact of moderate malnutrition on malaria pathology. Studies in female mice have typically shown a more moderate response to malaria infection, due to lower testosterone levels. As a result, female mice would not show similar differences between the well-nourished and moderate malnutrition after infection as our study was focused on determining the impact of malaria infection to the gut innate immunity during moderate malnutrition. Similarly, it has been shown that human males experience more severity with malarial disease [75].

Future studies will investigate the impact of the heightened inflammatory environment induced by malnutrition and malaria infection in the gut using a severe malnutrition diet. It has been suggested that nutrient deficiencies have the capacity to induce tight junction dysfunction and disorganization [61,76]. We foresee some sort of dysfunction in the tight junction proteins, in a severe malnutrition, which could involve deformities in occludin and (ZO)-1 expression. This deformities can lead to bacterial translocation to systemic tissues, which is often seen in malaria infection in the form of acute kidney injury [77]. 

Collectively, our results indicate decremented gut physiology induced by moderate malnutrition, which is exacerbated by malaria infection. Through utilization of a well-established moderately malnourished mouse model that mimics malnutrition seen in malaria endemic areas [34], we show the increasingly harmful effects of malaria and moderate malnutrition in the gut. This is shown by the increased proportion and number of activated macrophages, localized intestinal cytokine secretion, epithelial cell damage, and intestinal permeability. Future studies will explore deformities in the expression of tight junction proteins and possible bacterial translocation to precisely understand the mechanism by which malaria enhance detrimental effects of malnutrition in the gut.

## Figures and Tables

**Figure 1 nutrients-13-00913-f001:**
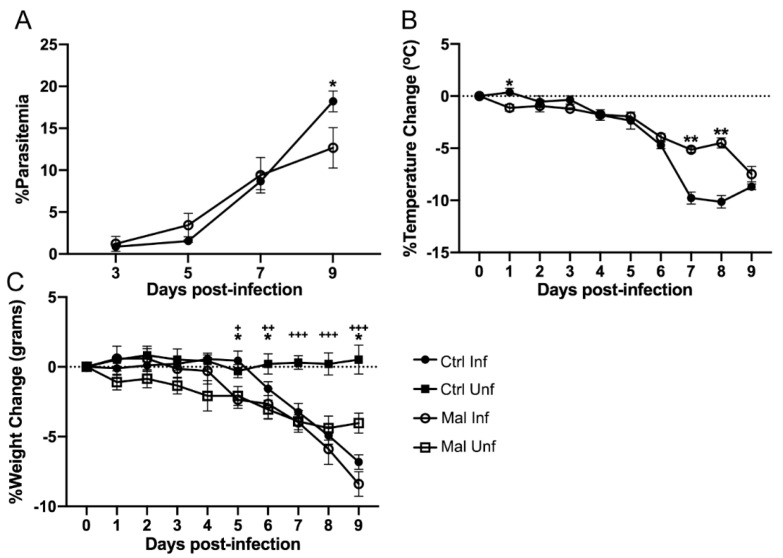
Parasite burden and pathological responses of control and moderately malnourished mice after infection with 1 × 10^5^
*P. chabaudi*. (**A**) Percent parasitemia. (**B**) Percent temperature change. (**C**) Percent change in body weight. All data are shown as means ± SE, *n* = 3 male mice per group, representative of 16 independent experiments. Statistical analysis was performed using a two-way ANOVA. +, * *p* < 0.05; ++, ** *p* < 0.01; +++ *p* < 0.001, indicates a statistically significant difference between treatments.

**Figure 2 nutrients-13-00913-f002:**
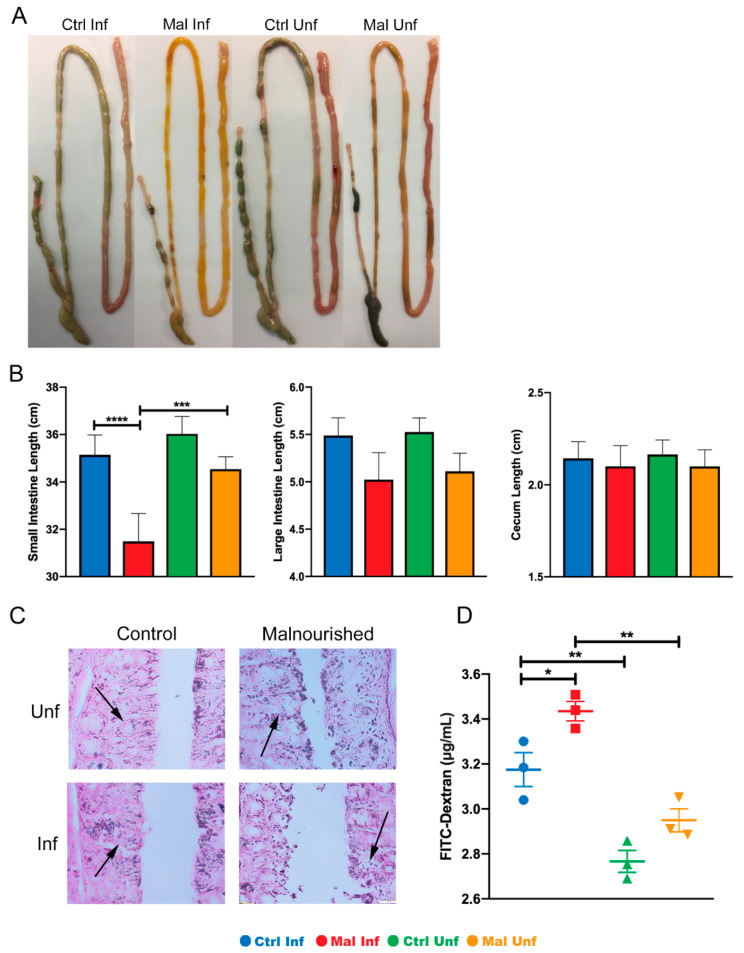
Moderate malnutrition alters gut phenotype, morphology and length after infection with *P. chabaudi*. (**A**) Macroscopic images of entire gut tissue. (**B**) Lengths of small intestines, large intestines, and cecum (average lengths from 16 independent experiments, *n* = 48). (**C**) Histology sections of the small intestine showing damaged mucosal cells indicated by arrows. (**D**) Serum FITC-Dextran concentrations. Data are shown as means ± SE. Histology and FITC-Dextran concentrations are representative of one experiment (*n* = 3). Statistical analysis was performed using a two-way ANOVA followed by the Tukey’s test. * *p* < 0.05; ** *p* <0.01; *** *p* < 0.001; **** *p* < 0.0001, indicates a statistically significant difference between treatments.

**Figure 3 nutrients-13-00913-f003:**
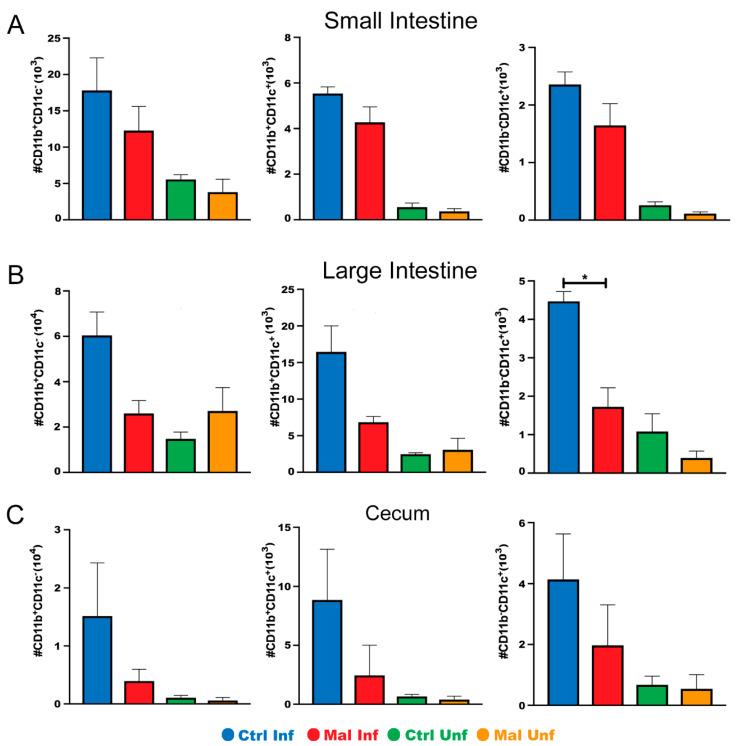
Moderately malnourished mice have lower numbers of innate immune cells in the gut during *P. chabaudi* infection. Flow cytometric analysis of surface staining for CD11b^+^ CD11c^-^, CD11b^+^ CD11c^+^, CD11b^−^ CD11c^+^ quantified cell numbers from the (**A**) small intestine, (**B**) large intestine, and (**C**) cecum of d9 *P. chabaudi* infected C57Bl/6 mice. Data are shown as means ± SE, *n* = 3 male mice per group, and is a representative of 3 independent experiments. Statistical analysis was performed using a two-way ANOVA followed by the Tukey’s test. * *p* < 0.05; indicates a statistically significant difference between treatments.

**Figure 4 nutrients-13-00913-f004:**
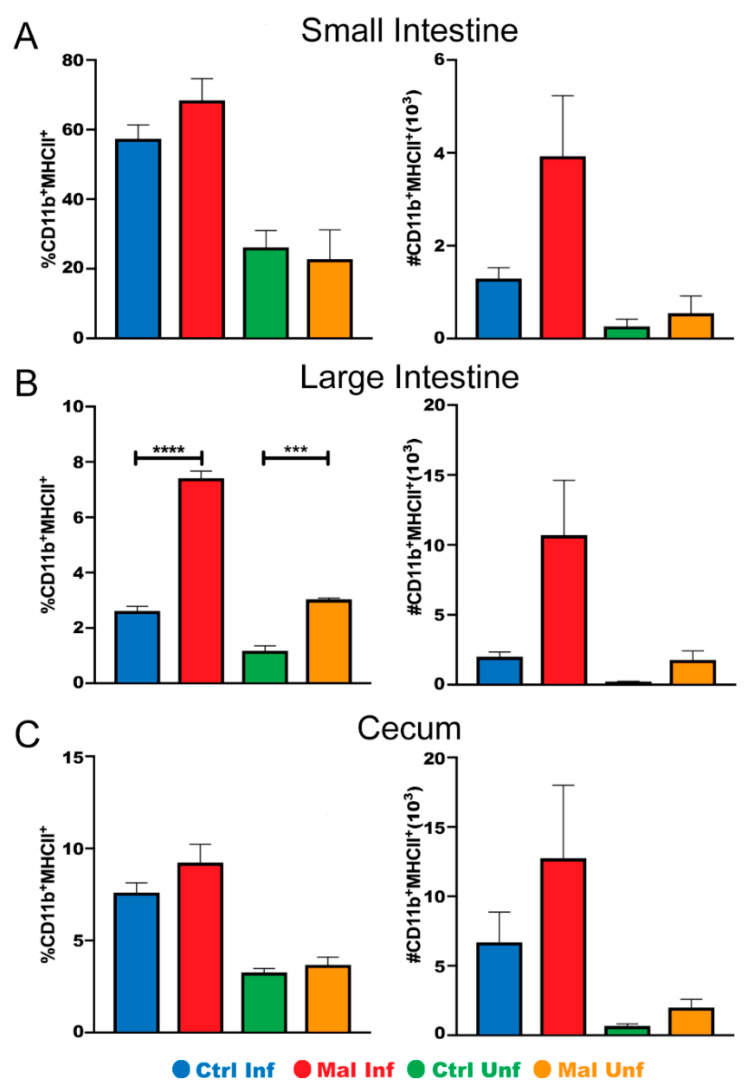
Moderately malnourished mice have increased proportions and numbers of activated macrophages in the gut during *P. chabaudi* infection. Representative flow cytometric analysis of surface staining for CD11b^+^/MHCII^+^ proportions and quantified cell numbers in the (**A**) small intestine, (**B**) large intestine, and (**C**) cecum at d9 *P. chabaudi* infection. Data are shown as means ± SE, *n* = 3 male mice per group, representative of 3 independent experiments. Statistical analysis was performed using a two-way ANOVA followed by Tukey’s test. *** *p* < 0.001; **** *p* < 0.0001; indicates a statistically significant difference between treatments.

**Figure 5 nutrients-13-00913-f005:**
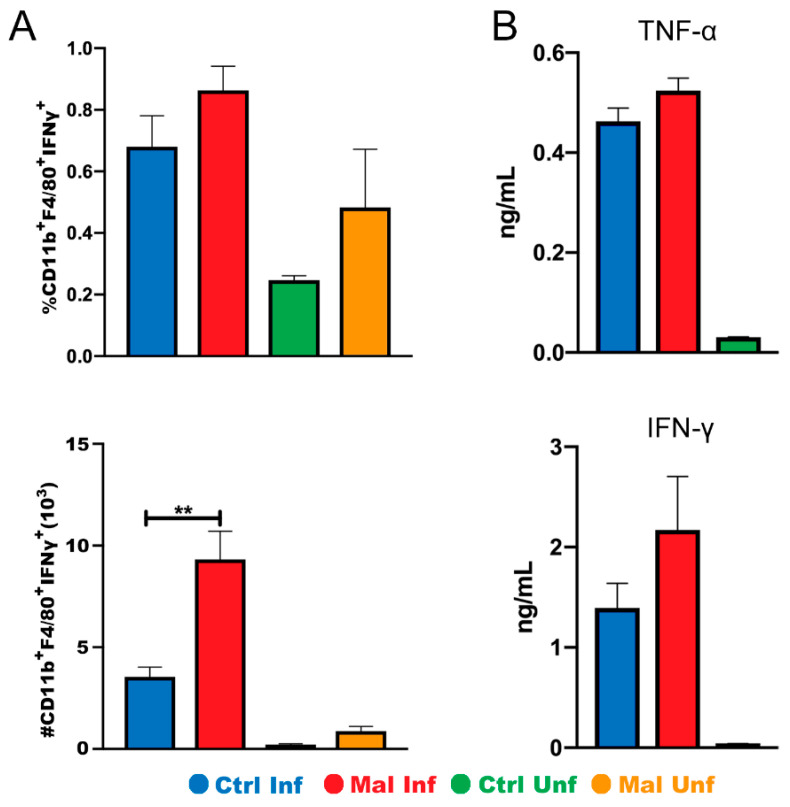
Moderate malnutrition increases IFN-γ secretion by macrophages in the small intestine but not systemically. (**A**) Graph showing flow cytometric analysis of intracellular staining for CD11b^+^MHCII^+^ IFNγ^+^ proportions (top) and quantified cell numbers (bottom). (**B**) Serum concentrations of TNF-α (top) and IFN-γ (bottom) at d9 *P. chabaudi* infection. Data for flow cytometry are shown as means ± SE, *n* = 3–4 male mice per group. Statistical analysis was performed using a two-way ANOVA. ** *p* < 0.01, indicates a statistically significant difference between treatments.

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
