# Peer review of "Plasmodium chabaudi Infection Alters Intestinal Morphology and Mucosal Innate Immunity in Moderately Malnourished Mice"

_nutrients, 2021, doi:10.3390/nu13030913_

Round 1

Reviewer 1 Report

The manuscript entitled: “Plasmodium chabaudi infection Alters Gut Integrity and Muco-2 sal Innate Immunity in Moderate Malnourished Mice” (nutrients-1083892) by Murr et al., aims to investigate the effects of malaria infection and malnutrition in gut.

This study fits perfectly into the “effects of malnutrition on malaria infection”, which has been the topic of much controversy over the years. In fact, and despite the contradictory results arisen from several studies, malnutrition has a direct link with malaria immunity and/or severity. The present study is a step forward as investigates the complex relationship between malaria and malnutrition and their effects on gut immunity and physiology. The aim is clear and of interest, however, the design/methods are not strong enough to extract clear and robust conclusions. More experiments are needed together with a better analysis of data.

MAJOR CONCERNS

INTRODUCTION SECTION

.- First lines: It is a bit confusing that the authors talk about micronutrient deficiency in the first sentence of the introduction but in the second line they talk about “protein” deficiency in children, which is a macronutrient.

.- Similarly, the authors do not explain why children suffering from PEM showed lower parasite burden during infection. This is caused, in most cases, by the glucose-6-phosphate deshidrogenase deficiency. This should be mentioned.

.- Along with reference 7, add: Shankar AH. Nutritional modulation of malaria morbidity and mortality. J Infect Dis. 2000;182:S37–53 . In this study the investigators found that controlled trials of vitamin A and zinc supplementation led to significant reduction in clinical malaria attacks.

.- Lines 45-54. Please, improve this paragraph to make it clearer to the readers. WHO recommends iron supplementation in conjunction with effective malaria prevention, however, it is also mentioned that iron deficiency may protect against malaria infection.

METHODOLOGY

.-This study is carried out just in male mice. Do the authors believe that these effects could be the same with females? This should be mentioned in the discussion section.

.- The authors should mention the number of animals per group. It is included in the figure legends but not in the methods. Please, clarify why only three animals are showed as representative of 16 independent experiments. An explanation concerning this issue should be added.

.-Why moderate malnutrition is used? This should be justified and include some references.  Similarly, it should be explained why only 3 grams of diet is provided per mouse and why they were not fed ad libitum.

.- Were fasted animals when sacrified? Please, mention

.- Statistical analyses are not appropriate and neither well explained in section 2.6. As two factors are being analyzed (infection and malnutrition) a two-factor Anova should be used. In this context, the potential interaction among both factors should be of great interest in this study.

.- Give details of the versions of software’s used for data analyses (Graph Pad, Excel).

.- Authors should explain the procedure for body temperature measurement.

RESULTS & DISCUSSION

.- Along this section, authors include some explanations and references (some examples: lines 170-172; 187-188…). From my point of view, this is not appropriate in the results section and should be moved to the introduction or discussion sections.

.- When talking about weight change, I guess it is body weight. Please specify.

.- Figure 2A. The results described related to this photograph should be corroborate by some histological analyses.

.- Figure 3: the authors mention (lines 224-225) that there were lower number of innate immune cell populations in all three sections of the gut in the moderate malnourished mice after infection, but statistical significance was only achieved in large intestine.

.- I have the feeling that the results are described only focused on malnutrition effects on infected animals, but the authors should make an effort in describing what happen after infection, what happen in malnourished mice and finally what happen in both cases. Thgis could strengthen the results obtained.

.- Despite the fact that a significant increased proportion and number of activate macrophages in gut are observed, this does not mean increased inflammation at systemic level. Thus, this should be investigated (levels of TNFa, IL6, IL1-beta in blood). Similarly, although the authors mention that the administration of FITC-Dextran is a method quite common to test intestinal permeability, this should be accompanied by some protein expression analyses of tight junction proteins. In the same line, bacterial translocation should also be determined.

.-What is the meaning of a shorter small intestine in the context of infection and malnutrition?

MINOR CONCERNS

.- Please, rpms should be indicated as gs

.- Correct the number of the section entitled: “Moderate malnutrition decrease mucosla innate immunity” as it should be 3.3 and not 3.2.

Author Response

We greatly appreciate your insightful review of of our manuscript as it helped improve the presentation and quality of our data. Please see the attachment document for response to the comments. We opted to upload a word document to allow us space to respond clearly to each comment.

Thanks

Authors

Reviewer 2 Report

Murr et al. explore the the complex relationship between malaria and malnutrition and their effects on gut immunity and physiology.

The authors reported the infected moderate malnourished mice had lower parasite burden at the peak of infection, but high levels of FITC-Dextran concentration in the blood serum, indicating increased intestinal permeability.

The objective of the paper is clear. Can the authors improve the manuscript , adding serum marker of gut  damage ? ( I-FABP , e-cadherin ). in order to fully measure gut damage.

It would  be interesting understand the role of microbial translocation (MT) in the described mechanism. Can the authors measure MT ? ( LPS-endocab-sCD14).

Author Response

The objective of the paper is clear. Can the authors improve the manuscript , adding serum marker of gut  damage ? ( I-FABP , e-cadherin ). in order to fully measure gut damage.

  • We appreciate this reviewer’s. Upon re-analysis of our data and based on gut phonotypes (new Figure 2c; page 7) and lower systemic cytokine secretion to the sera (new Figure 5b; page 10), when we used the moderate malnourished diet, we do not think that the damages to the gut were major to affect many mucosal proteins. In a different study using severe malnutrition, we are looking at the damage to tight junction protein including claudins and occluding.

It would  be interesting understand the role of microbial translocation (MT) in the described mechanism. Can the authors measure MT ? ( LPS-endocab-sCD14).

  • We do not think that there is microbial translocation with the moderate malnutrition. We have scaled down on our initial conclusions suggesting that there may be bacterial translocation. We have clarified this on page 12; lines 421-431. In a separate project using severe malnutrition we will determine MT.

Round 2

Reviewer 1 Report

From my point of view, the authors have carried out a remarkable effort in following the reviewer´s suggestions and the quality of the manuscript has significantly improved. For example, the new statistical analyses are more appropriate than the previous ones.

Author Response

Thanks for your review.